# Multifunctional Silver(I) Complexes with Metronidazole Drug Reveal Antimicrobial Properties and Antitumor Activity against Human Hepatoma and Colorectal Adenocarcinoma Cells [note 1]

**DOI:** 10.3390/cancers14040900

**Published:** 2022-02-11

**Authors:** Dominik Żyro, Lidia Radko, Agnieszka Śliwińska, Lilianna Chęcińska, Joachim Kusz, Izabela Korona-Głowniak, Agata Przekora, Michał Wójcik, Andrzej Posyniak, Justyn Ochocki

**Affiliations:** 1Department of Bioinorganic Chemistry, Medical University of Lodz, Muszyńskiego 1, 90-151 Łódź, Poland; 2Department of Pharmacology and Toxicology, National Veterinary Research Institute, Al. Partyzantów 57, 24-100 Puławy, Poland or lidia.radko@up.poznan.pl (L.R.); aposyn@piwet.pulawy.pl (A.P.); 3Department of Preclinical Sciences and Infectious Diseases, Poznan University of Life Sciences, Wołynska 35, 60-637 Poznan, Poland; 4Department of Nucleic Acids Biochemistry, Medical University of Łódź, Pomorska 251, 92-213 Łódź, Poland; agnieszka.sliwinska@umed.lodz.pl; 5Faculty of Chemistry, University of Lodz, Pomorska 163/165, 90-236 Łódź, Poland; lilianna.checinska@chemia.uni.lodz.pl; 6Institute of Physics, University of Silesia, 75 Pułku Piechoty 1, 41-500 Chorzów, Poland; joachim.kusz@us.edu.pl; 7Department of Pharmaceutical Microbiology, Medical University of Lublin, Chodźki 1, 20-093 Lublin, Poland; iza.glowniak@umlub.pl; 8Independent Unit of Tissue Engineering and Regenerative Medicine, Medical University of Lublin, Chodźki 1, 20-093 Lublin, Poland; agata.przekora@umlub.pl (A.P.); michal.wojcik@umlub.pl (M.W.)

**Keywords:** cytotoxicity, antimicrobial activity, silver(I) complexes, synthesis of silver(I) complexes, light stability, HepG2 cells, Caco-2 cells, Balb/c 3T3 cells

## Abstract

**Simple Summary:**

Our previous studies demonstrated that a silver(I) nitrate complex with metronidazole presented greater photo-stability, antimicrobial, cytotoxic and genotoxic properties than silver(I) nitrate. These advantages make the complex a better candidate for clinical therapy than pure salt. Therefore, in this study, we decided to synthetize and determine the chemical, cytotoxic and antimicrobial properties of [Ag(MTZ)_2_]_2_SO_4_, a novel metronidazole silver(I) complex, in comparison with pure salt Ag_2_SO_4_ and [Ag(MTZ)_2_NO_3_]. The photo-stability, cytotoxicity toward cancer cells and antimicrobial activity of [Ag(MTZ)_2_]_2_SO_4_ is higher than Ag_2_SO_4_. What is more, we found that the novel synthetized complex shows better cytotoxicity against cancer cells than [Ag(MTZ)_2_NO_3_]. Both complexes have similar biological activity against the majority of tested bacterial strains.

**Abstract:**

Silver salts and azole derivatives are well known for their antimicrobial properties. Recent evidence has demonstrated also their cytotoxic and genotoxic potential toward both normal and cancer cells. Still, little is known about the action of complexes of azoles with silver(I) salts. Thus, the goal of the study was to compare the chemical, cytotoxic and antimicrobial properties of metronidazole complexes with silver(I) nitrate and silver(I) sulfate to metronidazole and pure silver(I) salts. We synthetized a novel complex, [Ag(MTZ)_2_]_2_SO_4_, and confirmed its chemical structure and properties using ^1^H and ^13^C NMR spectroscopy and X-Ray, IR and elemental analysis. To establish the stability of complexes [Ag(MTZ)_2_NO_3_] and [Ag(MTZ)_2_]_2_SO_4_, they were exposed to daylight and UV-A rays and were visually assessed. Their cytotoxicity toward human cancer cells (HepG2, Caco-2) and mice normal fibroblasts (Balb/c 3T3 clone A31) was determined by MTT, NRU, TPC and LDH assays. The micro-dilution broth method was used to evaluate their antimicrobial properties against Gram-positive and Gram-negative bacteria. A biofilm eradication study was also performed using the crystal violet method and confocal laser scanning microscopy. The photo-stability of the complexes was higher than silver(I) salts. In human cancer cells, [Ag(MTZ)_2_]_2_SO_4_ was more cytotoxic than Ag_2_SO_4_ and, in turn, AgNO_3_ was more cytotoxic than [Ag(MTZ)_2_NO_3_]. For Balb/c 3T3 cells, Ag_2_SO_4_ was more cytotoxic than [Ag(MTZ)_2_]_2_SO_4_, while the cytotoxicity of AgNO_3_ and [Ag(MTZ)_2_NO_3_] was similar. Metronidazole in the tested concentration range was non-cytotoxic for both normal and cancer cells. The complexes showed increased bioactivity against aerobic and facultative anaerobic bacteria when compared to metronidazole. For the majority of the tested bacterial strains, the silver(I) salts and complexes showed a higher antibacterial activity than MTZ; however, some bacterial strains presented the reverse effect. Our results showed that silver(I) complexes present higher photo-stability, cytotoxicity and antimicrobial activity in comparison to MTZ and, to a certain extent, to silver(I) salts.

## 1. Introduction

Silver(I) salts have well-documented antimicrobial properties. The mechanism of the bactericidal action of silver compounds is based on their binding to the thiol groups (-SH) of proteins, including bacterial enzymes, leading to their inactivation by the formation of sparingly soluble connections with cytoplasmic proteins [1]. As a result, a protective coat is created that prevents pathogens from penetrating into the cell. Silver nitrate (Argenti nitras, Lapis) is one of the most commonly used silver(I) salts because of its germicidal and bacteriostatic properties. To prevent gonococcal conjunctivitis in newborns, silver(I) nitrate is employed using Credé’s procedure (Mova Nitrat Pipette^®^, Teva Pharmaceuticals Poland, Warsaw, Poland) [2]. This salt in higher concentrations is also used in otolaryngology in the form of a solution or stick for a chemical cauterization procedure, which consists of coagulating a pathological or excessively bleeding lesion to accelerate the healing process. Cumulating in vitro evidence reveals the cytotoxic and genotoxic potential of the silver(I) salts toward both normal and cancer cells [3,4].

Metronidazole (MTZ) is a 5-nitroimidazole derivative (Figure 1). It acts primarily on anaerobic organisms, namely, bacteria and protozoa. It easily penetrates their cells. The antimicrobial mechanism action involves the mediated reduction of the nitro group (-NO_2_) of MTZ to the amino group (-NH_2_) by ferredoxin (present in anaerobic organisms). The resulting product causes DNA damage to the microbes, leading to their cell death. Metronidazole was considered to be anti-tumor active due to its affinity for penetration and accumulation in hypoxic tumors [5]. Nitroimidazoles can act as oxygen mimics, especially in hypoxic cells, so they can be used during irradiation to fix the radiation-induced damage in DNA or other macromolecules [6]. It was also reported that some derivatives of 2-nitroimidazole inhibited tumor-specific angiogenesis by blocking the production of angiogenic factors [7]. Metronidazole is readily absorbed after oral administration, and it is also used topically on the skin and vaginally [8,9].

Silver, a d-block metal, has the ability to form complex compounds. It was demonstrated that silver(I) complexes exhibit more varied chemical properties than pure silver(I) salts [3,4,10]. Moreover, numerous studies have revealed that silver(I) complexes possess better biological activity than the free ligand, including cytotoxicity, genotoxicity, and antimicrobial activity [11,12,13,14]. Among silver(I) complexes, the best explored are complexes with nitrate counter-ions with azole and pyridine derivative ligands. However, nothing is known about complexes of silver(I) sulfate comprising these ligands. Ag_2_SO_4_ has not been applicated in medicine due to a low degree of ionization and resulting poor solubility. Therefore, it is tempting to investigate whether a complex of silver(I) sulfate with azole or pyridine derivative ligands have different chemical and biological properties. To address this problem, we synthetized a novel complex, [Ag(MTZ)_2_]_2_SO_4_, and examined its chemical structure and properties using ^1^H and ^13^C NMR spectroscopy and X-ray, infrared radiation (IR) and elemental analysis. Therefore, the aim of the study was to compare the chemical, cytotoxic and microbiological properties of metronidazole complexes with silver(I) nitrate and silver(I) sulfate to metronidazole and pure silver(I) salts.

## 2. Materials and Methods

### 2.1. Chemicals and Reagents

Metronidazole (CAS: 443-48-1; molecular weight: 171.16 g/mol) was purchased from Alfa Aesar (Kandel, Germany). Silver nitrate (CAS: 7761-88-8; molecular weight: 169.87 g/mol), silver sulfate (CAS: 10294-26-5; molecular weight: 311.79 g/mol) and cisplatin (CAS: 15663-27-1; molecular weight: 300.05 g/mol) were purchased from Sigma-Aldrich (Poznań, Poland). Triton X-100, dimethyl sulfoxide (DMSO), fetal bovine serum (FBS), bovine calf serum (BCS), neutral red dye (NR), Coomassie brilliant blue R-250 dye, 3-(4,5-dimethylthiazol-2-yl)-2,5-diphenyltetrazolium bromide (MTT), trypsin-EDTA and an antibiotic solution (10,000 U/mL of penicillin, 10 mg/mL of streptomycin) were purchased from Sigma-Aldrich (Poznań, Poland). All other chemicals were purchased from commercial suppliers and were of the highest available purity. 

### 2.2. Synthetic Procedures

#### 2.2.1. Synthesis of [Ag(MTZ)_2_NO_3_]

We described a modified and simplified synthesis of this complex in our earlier papers as a “one-step method” [4]. Specifically, we prepared a solution of AgNO_3_ (0.855 g, 5 mmol) in water (25 mL) and added 1.71 g (10 mmol) of metronidazole. The reaction mixture was stirred at 80 °C for 2 min. After cooling to an ambient temperature, the resulting white crystalline solid (needles) of the complex was filtered, washed twice with diethyl ether and air-dried. Molecular weight: 512.18 g/mol; yield: 2.3 g (89.67%). The elemental analysis measured (calculated %): C 28.18 (28.14), H 3.29 (3.54), N 19.13 (19.14); melting point: 151–153 °C. ^1^H NMR (600 MHz, DMSO): δ = 2.51 ppm (s, 6H, 2 × CH_3_), 3.71 (q, 4H, 2 × CH_2_-O, *J* = 5.5 Hz), 4.39 (t, 4H, 2 × CH_2_-N), 5.05 (t, 2H, 2 × –OH, *J* = 5.5 Hz), 8.07 (s, 2H, 2 × CH-N). ^1^H NMR (600 MHz, DMSO + D_2_O): δ = 2.51 ppm (s, 6H, 2 × CH_3_), 3.69 (t, 4H, 2 × CH_2_-O, *J* = 5.5 Hz), 4.38 (t, 4H, 2 × CH_2_-N), 8.06 (s, 2H, 2 × CH-N). IR (KBr) *v* _max_ (cm^−1^): (O-H, C-H) 3214, 3101, 3061, 3018, 2930, 2870; (C=N, C=C) 1555; (NO_2_, C-H) 1467, (NO_3_^−^, NO_2_, C-H) 1385, 1341; (C-O) 1080; (NO_3_^−^) 800.

#### 2.2.2. Synthesis of [Ag(MTZ)_2_]_2_SO_4_·5H_2_O

Here, a 0.685 g (4 mmol) portion of metronidazole was added to a solution of Ag_2_SO_4_ (0.312 g, 1 mmol) in 35 mL of hot water (90 °C). Next, the whole reaction mixture was stirred for 15 min. at 80–90 °C. The solution was allowed to crystallize, and crystals were obtained in few days at fridge temperature (2–8 °C) (see the Appendix A). The crystals were collected, washed twice with diethyl ether and air-dried. Molecular weight: 1086.43 g/mol; yield: 0.9 g of hydrated salt of the complex (82.84%). Elemental analysis for C_24_H_46_N_12_O_21_Ag_2_S calculated (found): C 26.52 (26.43)%, H 4.27 (4.00)%, N 15.47 (15.48)%. Melting point: 168–171 °C (decomposition); crystallization water is released in 79–82 °C and partially dissolves the complex. ^1^H NMR (600 MHz, DMSO): δ = 2.47 ppm (s, 12H, 4 × CH_3_), 3.69 (q, 8H, 4 × CH_2_-O, *J* = 5.09 Hz), 4.36 (t, 8H, 4 × CH_2_-N), 5.03 (t, 4H, 4 × OH, *J* = 2.5 Hz), 8.04 (s, 4H, 4 × CH-N). ^1^H NMR (600 MHz, DMSO + D_2_O): δ = 2.45 ppm (s, 12H, 4 × CH_3_), 3.67 (m, 8H, 4 × CH_2_-O), 4.35 (t, 8H, 4 × CH_2_-N, *J* = 5.7 Hz), 8.02 (s, 4H, 4 × CH-N). ^13^C NMR (600 MHz, DMSO-d_6_): δ 152.42 (N=C-N), 133.45 (CH-N), 60.21 (CH_2_-O), 48.74 (N-CH_2_), 14.74 (CH_3_-imidazole). IR (KBr) *v*_max_ (cm^−1^): (O-H, C-H) 3220, 3100, 3017, 2981, 2957, 2850; (C=N, C=C) 1536; (NO_2_, C-H) 1487, (NO_2_, C-H) 1368, 1354; (C-O) 1073; (SO_4_^2−^) 1115. ^1^H NMR analysis shows that the proton from the OH group (δ = 5.03 ppm) of metronidazole remains in the structure of the complex. The proton is exchanged after being dissolved in deuterium water. The ^1^H NMR and ^13^C NMR reproduced spectra are provided in the Appendix A. 

### 2.3. Light Stability of Silver(I) Complexes

The light stability of the [Ag(MTZ)_2_NO_3_] and [Ag(MTZ)_2_]_2_SO_4_∙5H_2_O complexes, as well as silver(I) salts, was studied in normal daylight in the air atmosphere, in UV-A light and in the dark (see Appendix A). The complexes and salts were dissolved in water. The concentration of each solution was 0.025 mol/L. For comparative purposes, the solutions of the complexes and silver(I) salts were applied onto tissue paper and paper in the amount of 1.25 µmol (50 µL of the solution). Stability in normal daylight and in the dark was monitored visually for 48 h, with image documentations after 0, 10, 30, 60, 90, 120 and 180 min., and after 24 and 48 h. Stability in UV-A was monitored visually for 1 h, with image documentation after 0, 1, 2, 5, 10, 20, 30, 60, 120, 300, 600, 1800 and 3600 s from the application. The images of samples are shown in the Appendix A.

### 2.4. X-ray Diffraction Study

Crystal data for [Ag(C_6_H_9_N_3_O_3_)_2_]_2_SO_4_·5H_2_O (*M* = 1086.53 g/mol): triclinic, space group *P*$\bar{1}$ (No. 2), *a* = 11.2281(3) Å, *b* = 12.3371(5) Å, *c* = 16.9502(7) Å, α = 97.762(3)°, β = 108.111(3)°, γ = 111.720(3)°, *V* = 1988.31(14) Å^3^, *Z* = 2, *T* = 150(2) K, μ(MoKα) = 0.134 mm^−1^, *D*_calc_ = 1.815 g/cm^3^, 20765 reflections were measured (3.1° ≤ Θ ≤ 29.0°), 10523 were unique (*R*_int_ = 0.026). They were used in all calculations. The final *R*1 was 0.039 [*I* > 2σ(*I*)] and *wR*2 was 0.109 (all data). 

Single crystal X-ray data were collected on a SuperNova diffractometer with an Atlas detector using MoKα at 150 K. The data reduction was performed with CrysAlis PRO software [15]. The structure was solved with SHELXT [16] and refined with SHELXL [17]. (C)-H atoms were added at the geometrically idealized positions and constrained to their parent atoms as a rigid body. (O)-H atoms in metronidazole ligands were found in a difference map and refined isotropically. The crystal structure contains 5 water molecules (O100, O200, O300, O400 and O500) for which all required H-atoms were not included in the model probably because of their high disorder. A Hirshfeld surface analysis of the metallic centre Ag1 and Ag2 was performed using CrystalExplorer [18,19]. Supplementary crystallographic data for this paper can be found at the Cambridge Crystallographic Data Center under the depository number CCDC-2048084.

### 2.5. Chemistry

The ^1^H and ^13^C NMR spectra were recorded at the Faculty of Chemistry (University of Lodz, Lodz, Poland) with a Bruker Advance III 600 MHz spectrophotometer (Bruker Corporation, Billerica, MA, USA) at room temperature. IR spectra were recorded at the Faculty of Pharmacy (Medical University of Lodz, Lodz, Poland) in the range of 4000–400 cm^−1^ on a Bruker Alpha-T (Bruker Corporation, Billerica, MA, USA) using potassium bromide. Elemental analyses (C, H, N) were performed at the Medical University of Lodz (Faculty of Pharmacy, Lodz, Poland) with a Perkin Elmer 2400 analyzer (PerkinElmer, Waltham, MA, USA). The melting points of all the compounds were determined with Böethius apparatus (Franz Küstner Nachf. KG, HMK, Dresden, Germany). A lamp with a wavelength of 365 nm was used as the source of UV-A rays (Vilber Lourmat, Marne-la-Valée, France).

### 2.6. Cell Culture and Cytotoxicity Assessment

#### 2.6.1. Cell Line Cultures

The human hepatoma (HepG2) and colorectal adenocarcinoma (Caco-2) cells were purchased from the American Type Culture Collection: ATCC HB-8065 and ATCC HTB-37, respectively. These cells were cultured in Minimum Essential Medium Eagle (MEME) (ATCC). The murine fibroblasts cell line (Balb/c 3T3 clone A31) (gift from the Department of Swine Diseases of the National Veterinary Research Institute in Pulawy) was cultured in Dulbecco’s Modified Eagle Medium (DMEM) (ATCC). The media were supplemented with 10% BCS (Balb/c 3T3), 10% FBS (HepG2), 20% FBS (Caco-2), 1% L-glutamine, 1% antibiotic solution. The cells were maintained in 75 cm^2^ cell culture flasks (NUNC) in a humidified incubator at 37 °C in an atmosphere of 5% CO_2_. The medium was refreshed every two or three days, and the cells were trypsinized by 0.25% trypsin–0.02% EDTA after reaching 70–80% confluence. Single cell suspensions were prepared and adjusted to a density of 2 × 10^5^ cell/mL^−1^ (HepG2), 1 × 10^5^ cell/mL^−1^ (Caco-2) and 5 × 10^4^ cell/mL^−1^ (Balb/c 3T3). The cell suspension was transferred to 96-well plates (100 µL/well) and incubated for 24 h before the exposure to the studied compounds. Stock solutions of ligands and metal complexes were prepared in sterile water and diluted with culture medium to obtain a concentration range from 0.1 to 10 µM. The final concentration of sterile water was 0.1% in the medium. The medium used for test solutions and in control preparation did not contain serum and antibiotics. As a negative control, cultured cells were grown in the absence of studied compounds. Each concentration was tested in six replicates in three independent experiments. Cytotoxicity was assessed after 72 h of exposure of the cells to compounds. The medium was not changed during the incubation time.

#### 2.6.2. MTT Assay

The metabolic activity of living cells was assessed by the measurement of the activity of dehydrogenases [20]. After the incubation of the cells with the studied compounds, 10 μL of the MTT solution (5 mg/mL in PBS) was added to each well of 96-well plates and incubated. After 3 h, the MTT solution was removed, and the intracellular formazan crystals were dissolved in 100 µL DMSO. The plate was shaken for 15 min at room temperature and transferred to a microplate reader Synergy HTX multi-mode reader (BioTek^®^ Instruments Inc., Winooski, VT, USA) to measure the absorbance at 570 nm, using a blank as a reference. Cytotoxicity was expressed as a percentage of the negative control [4].

#### 2.6.3. NRU Assay

This method is based on staining living cells with neutral red, which readily diffuses through the plasma membrane and accumulates in lysosomes [21]. After the incubation, the medium containing the tested substance was removed and the cells were washed with PBS. Then, 100 µL/well of NR solution (50 µg/mL) was added for 3 h. After this time, the cells were washed with PBS. The dye from viable cells was released by extraction with a mixture of acetic acid, ethanol and water (1:50:49, v:v:v). After 10 min of shaking, the absorbance of the dissolved NR was measured using a Synergy HTX multi-mode reader (BioTek^®^ Instruments Inc., Winooski, VT, USA) at 540 nm using a blank as a reference. Cytotoxicity was expressed as a percentage of the negative control [4].

#### 2.6.4. TPC Assay

This assay is based upon staining the total cellular protein (proliferation) [22]. After the incubation, the medium containing the drug was removed, and 100 µL Coomassie brilliant blue R-250 dye was added to each well. The plate was shaken for 10 min. Then, the stain was removed and the cells were rinsed twice with 100 µL of washing solution (glacial acetic acid/ethanol/water, 5:10:85, v:v:v). After that, 100 µL of the desorbing solution (1 M potassium acetate) was added, and plates were shaken again for 10 min. The absorbance was measured at 595 nm in a microplate reader Synergy HTX multi-mode reader (BioTek^®^ Instruments Inc., Winooski, VT, USA) using a blank as a reference. Cytotoxicity was expressed as a percentage of the negative control [4].

#### 2.6.5. LDH Leakage Assay

The integrity of the plasma membrane was assessed through the test of lactate dehydrogenase (LDH) release [23], which was monitored using the commercially available Cytotoxicity Detection Kit (LDH) (Roche Diagnostics, Warsaw, Poland). The medium (100 µL/well) without cells was transferred into the corresponding wells of an optically clear 96-well flat bottom microplate, and 100 µL reaction mixture was added to each well. Then, the plates were incubated for 30 min at room temperature in darkness. After that time, 50 µL/well of 1 M HCl was added to stop the reaction. The absorbance was measured at 492 nm in a microplate reader Synergy HTX multi-mode reader (BioTek^®^ Instruments Inc., Winooski, VT, USA) using a blank as a reference [4].

#### 2.6.6. Statistical Analysis

The study was performed in three independent experiments. The obtained results are presented as mean values ± SD (standard deviation). The assessment of the cytotoxicity data used a one-way analysis of variance (ANOVA) followed by Dunnett’s post hoc test, which was applied to determine significance relative to the negative control. The cytotoxicity concentration (IC_50_) necessary for a 50% inhibition of cell viability by the drug was calculated using GraphPad Prism 5.0. Statistical comparisons among IC_50_ values were performed with the analysis of variance (ANOVA) followed by the Tukey test, and differences were considered to be statistically significant at *p* ≤ 0.05.

### 2.7. Antimicrobial Activity

Ag complexes were screened for antibacterial activities using the micro-dilution broth method according to the European Committee on Antimicrobial Susceptibility Testing (EUCAST) (www.eucast.org, accessed on 1 March 2021) using Mueller–Hinton broth and Mueller–Hinton broth with 5% lysed horse blood for the growth of non-fastidious and fastidious bacteria. The minimal inhibitory concentration (MIC) and minimal bactericidal concentration (MBC) of the tested complexes were evaluated with the panel of the reference aerobic and anaerobic microorganisms from the American Type Culture Collection (ATCC), including Gram-negative bacteria (*Escherichia coli* ATCC 25922, *Salmonella* Typhimurium ATCC 14028, *Klebsiella pneumoniae* ATCC 13883, *Pseudomonas aeruginosa* ATCC 9027, *Proteus mirabilis* ATCC 12453), *Bacteroides fragilis* ATCC 10240, *Fusobacterium nucleatum* ATCC 25586, *Veillonella parvula* ATCC 10790 and Gram-positive bacteria (*Staphylococcus aureus* ATCC 25923, *Staphylococcus epidermidis* ATCC 12228, *Micrococcus luteus* ATCC 10240, *Enterococcus faecalis* ATCC 29212, *Bacillus subtilis* ATCC 6633, *Bacillus cereus* ATCC 10876), *Actinomyces israelii* ATCC 10049, *Propionibacterium acnes* ATCC 11827 and *Clostridium perfringens* ATCC 13124. The antimicrobial assays were performed in the same manner as in our previous research [24]. The stability of the Ag complexes’ activity was confirmed by antimicrobial activity testing after 30 min of UV exposition. Each experiment was repeated in triplicate. Representative data are presented. 

### 2.8. Biofilm Eradication Study

Biofilm eradication was assayed in 96-well flat-bottom polystyrene plates where a culture of bacterial inoculum was added to reach a final concentration of 5 × 10^5^ CFU/m, with a final volume of 100 μL per well. The biofilms of *S. aureus* ATCC 25923 and *E. coli* ATCC 25922 were cultured on polystyrene plates for 1 day. Complexes were two-fold serially diluted in broth, and then, after overnight incubation, the supernatant was removed and 100 μL of medium with two-fold serial dilutions of compounds was added to each well. Plates were incubated for another 24 h at 37 °C. A control was defined as bacteria in the absence of compounds.

### 2.9. Crystal Violet (CV) Method

To determine the amount of biofilm after compound incubation, the supernatant was gently removed, and the formed biofilms were washed twice with 100 μL of saline solution to withdraw planktonic cells. The remaining biofilm was fixed with 100 μL of ethanol for 15 min, and then stained with 100 μL of crystal violet (CV) 1% (v/v) for 10 min. The dye was removed, wells were washed twice with 200 μL of distilled water and the plate was dried at 37 °C for 30 min. Finally, 100 μL of ethanol 96% was added, samples were homogenized by gentle agitation, and the absorbance was measured in a microplate reader at 570 nm. The OD_570_ of the alcohol–dye solution in each well was read by using a microplate reader (BioTek ELx800). The blank control wells, without or with a two-fold dilution of the tested compounds and reference agents added to the same broth, but without bacterial suspension, were incubated under the same conditions. OD_570_ values read in these wells were the ODc values, being the reference point for determining biofilm mass value.

### 2.10. Qualitative Determination of Antibiofilm Activity via Confocal Laser Scanning Microscope (CLSM)

A Viability/Cytotoxicity Assay kit for Bacteria Live and Dead Cells (Biotium, Fremont, CA, USA) was used to qualitatively determine the antibiofilm properties of the compounds. This kit consists of two DNA dyes, DMAO and EthD-III. DMAO has the ability to stain both live and dead bacteria, whereas EthD-III dye stains only the DNA of dead cells. The application of a DMAO/EthD-III mixture enabled us to distinguish the viable bacteria from the dead cells. Bacteria with intact cell membranes exhibit pure green fluorescence, whereas cells with damaged membranes reveal red or yellow (the overlapping of two colors) fluorescence. After 24 h exposure of bacteria to different concentrations of the compounds, the biofilms formed on the polystyrene wells were gently rinsed with phosphate-buffered saline (PBS, Pan-Biotech, Aidenbach, Germany) to remove planktonic bacteria. Then, the samples were stained using a DMAO/EthD-III mixture according to the manufacturer’s protocol. The bacterial biofilm was observed using a confocal laser scanning microscope (CLSM, Olympus Fluoview equipped with FV1000, Olympus, Tokyo, Japan).

## 3. Results

### 3.1. Synthesis of Silver(I) Complexes

#### 3.1.1. Synthesis of [Ag(MTZ)_2_NO_3_]

The modified synthesis of the complex was performed earlier [4]. Water was used as a solvent instead of a water/ethanol solution. The silver(I) complex of metronidazole was synthesized in an easy one-step process through the reaction of AgNO_3_ with metronidazole (1:2) (Figure 1) in water. The complex was obtained with a good yield and purity (see the Materials and Methods Section). 

#### 3.1.2. Synthesis of [Ag(MTZ)_2_]_2_SO_4_·5H_2_O

The synthesis of the complex was carried out in an aqueous environment. The metronidazole silver(I) complex was synthesized in an easy, one-pot process by reacting Ag_2_SO_4_ with metronidazole (1:2) (Figure 2). The complex was obtained with a good yield and purity (see Materials and Methods Section).

### 3.2. Light Stability of Siver(I) Complexes

Ag_2_SO_4_ and AgNO_3_ began to darken after 10 min of exposure on the paper and after 30 min on the tissue paper. Ag_2_SO_4_ and AgNO_3_ darkened completely after 24 h on both groundworks. The complexes with sulfate and nitrate began to darken after 30 min of exposure on the paper and after 60 min on the tissue paper. The process of decomposition of the complexes proceeded quite slowly, and the complexes completely darkened after about 2 days, decomposing the fastest on paper. (Appendix A in the dark, Appendix A in the light; see Appendix A). The action of UV-A rays accelerated the process of decomposition of the silver(I) complexes to a large extent, with practically complete disintegration occurring after 5 min. (Appendix A in UV-A; see Appendix A). Pure salts were decomposed after 2 min in the UV-A rays. The complexes of metronidazole with silver(I) nitrate and silver(I) sulfate show better stability and resistance to UV-A radiation, while the pure salts are less stable. Comparatively, the effect of light and UV-A radiation affects only the substances applied onto the surfaces that they can permeate, while the aqueous solutions are quite stable, and changes only occur after the natural evaporation of the solvent.

### 3.3. Crystal Structure of [Ag(MTZ)_2_]_2_SO_4_·5H_2_O

The novel complex crystallized in a triclinic system in the *P*$\bar{1}$ space group, Z = 2. The asymmetric unit is presented in Figure 2, and consists of two silver ions, four metronidazole ligands, one sulfate counter ion and five water molecules. Both silver cations adopt a four-coordinate sawhorse geometry with two Ag-N bonds (2.2 Å, Table 1) and two relatively short Ag-O bonds with the distance of 2.6 Å. A similar coordination scheme of the metal center was described for the Ag(I) complexes of 1,3,5-tris(4-cyanobenzoyl)benzene with OTf and PF_6_ counter-ions [25]. Additionally, two much weaker Ag···O interactions of 2.9 Å (still shorter than the sum of the van der Waals radii of Ag and O; 3.24 Å) are observed, completing the coordination around silver cations to octahedron. Such a type of weak Ag···O contact is rarely discussed in the literature; however, they can be easily found with a closer inspection of the crystal structures. The Ag coordination scheme is presented in Figure 3 and confirmed by the Hirshfeld surface of the metallic center of Ag1 and Ag2 (Figure 4). The axial sites of each octahedron are occupied by nitrogen atoms, whereas the basal plane is formed by four oxygen atoms. The normalized contact distance (d_norm_) encoded on the Hirshfeld surface indicates the strength of the interaction (close contacts are shown as red regions, weaker interactions as white and no contacts as blue) [26]. It is clearly seen that the longer interactions are still visible on the Hirshfeld surface of each Ag center; however, they are much weaker than others, as expected.

The two silver ions, arranged alternately, form a 1D-polymeric chain running along the [−1–10] direction. The adjacent chains are linked into 2D-polymeric layer through the weakest Ag···O contacts, which is illustrated in Figure 5. There is a space filled by non-coordinated sulfate counter-ions and water molecules between the layers built from silver(I) cations and metronidazole ligands. On the basis of calculations in PLATON [27], such a solvent-accessible void space occupies 22% of the total crystal volume. Table 2 presents the geometry of the O-H···O hydrogen bonds formed between the hydroxyl group of the metronidazole ligands and the oxygen atoms from the water molecules or sulfate anions. The complete analysis of the hydrogen-bonding interactions is not possible because of the omission of the hydrogen atoms from the water molecules in the refinement.

### 3.4. Cytotoxicity of the Compounds

The cytotoxic activity of the novel synthetized compounds [Ag(MTZ)_2_]_2_SO_4_ and [Ag(MTZ)_2_NO_3_] in comparison to MTZ, Ag_2_SO_4_ and AgNO_3_ was investigated for human hepatocellular carcinoma HepG2 cells, human colorectal adenocarcinoma Caco-2 cells and non-cancer murine Balb/c 3T3 fibroblasts using four biochemical assays: mitochondrial (the 3-(4,5-dimethyl-2-thiazolyl)-2,5-diphenyl-2-H-tetrazolium bromide (MTT) assay) activity, lysosomal (neutral red uptake (NRU) assay) activity, total protein content (TPC assay) and cellular membrane integrity (lactate dehydrogenase (LDH) assay) after 72 h of exposition, as presented in Figure 6. The cytotoxic potential of studied compounds expressed as half-maximal inhibitory concentration (IC_50_) values are presented in Table 3.

The results indicated that human carcinoma (HepG2 and Caco-2) cells were more sensitive to novel synthetized compound [Ag(MTZ)_2_]_2_SO_4_ than non-cancer Balb/c 3T3 fibroblasts (Table 3). The obtained cytotoxic concentrations (IC_50_) values shown that the studied complex [Ag(MTZ)_2_]_2_SO_4_ (2.63–4.63 µM) is two times more toxic for carcinoma cells than [Ag(MTZ)_2_NO_3_] (4.12–10 µM) (Table 3). These values were lower compared to the IC_50_ values for cisplatin (<0.1–5.20 µM) on studied cells (Table 3). It is worth noting that [Ag(MTZ)_2_]_2_SO_4_ strongly inhibited mitochondrial, lysosomal and proliferative activity in HepG2 cells and all endpoints in Caco-2 cells compared to Ag_2_SO_4_ (Table 3). However, the opposite situation was observed for non-cancer Balb/c 3T3 cells. Ag_2_SO_4_ was two-fold more toxic (1.92–2.57 µM) than the compound [Ag(MTZ)_2_]_2_]SO_4_ (4.06–4.53 µM). When analyzing the results of the decrease in the viability of the cells presented in Figure 6, we noticed that Caco-2 cells were more sensitive to the compound than HepG2 and Balb/c 3T3 cells. The silver(I) complex significantly (*p* ≤ 0.05) inhibited effects in all used endpoints in HepG2 and Balb/c 3T3 cells, and this was observed starting from the concentration of 5.0 μM (Figure 6). In the case of Caco-2 cells, the significant (*p* ≤ 0.05) disintegration of cellular membranes at the concentration of 0.5 μM, was recorded (Figure 6 Caco-2: LDH). At a concentration of 1.0 μM, the inhibition of mitochondrial and lysosomal activities was observed (Figure 6 Caco-2: MTT, NRU). At higher concentration of 5.0 μM, the silver complex decreased the total protein content in the cell cultures (Figure 6, Caco-2: TPC).

The combination of metronidazole with AgNO_3_ resulted in the non-cancer Balb/c 3T3 fibroblasts being more sensitive to [Ag(MTZ)_2_NO_3_] than human carcinoma (HepG2 and Caco-2) cells (Table 3). Our earlier article describes the cytotoxic effects of this compound on HepG2 and Balb/c 3T3 cells (the solvent of compounds was DMSO; in this study, the solvent was sterile water) [4]. 

There were no differences between the results (IC_50_ values) for the studied compounds obtained in both studies. The inhibited lysosomal and proliferation activity in Caco2 cells were stronger than in HepG2 cells. It should also be noted that [Ag(MTZ)_2_NO_3_] was more toxic compared to AgNO_3_ in cancer HepG2 and Caco-2 cells. The opposite situation was observed for [Ag(MTZ)_2_]_2_SO_4_ and Ag_2_SO_4_. The IC_50_ values for all studied endpoints were higher compared to IC_50_ values for cisplatin on HepG2 cells (Table 3). However, the inhibition of mitochondrial activity (MTT, 7.32 μM) and changes to the cellular membrane (LDH, 4.42 μM) in Caco-2 cells were more strongly exhibited than the activity of cisplatin (9.12 and >10 μM, respectively). Analyzing the results of the decrease in the viability of the cells presented in Figure 6, we noticed that Caco-2 cells were as sensitive to the compound as HepG2 cells (in the MTT, NRU and TPC assays), and less sensitive to [Ag(MTZ)_2_]_2_SO_4_ activity. The studied silver(I) complex at a higher concentration (1.0 μM) than for [Ag(MTZ)_2_SO_4_] significantly (*p* ≤ 0.05) affected cell membranes, leading to disintegration (Figure 6 Caco-2: LDH). At a concentration of 5.0 μM, inhibited mitochondrial and lysosomal activities were observed (Figure 6, Caco-2: MTT, NRU). At a higher concentration of 10.0 μM, this silver complex decreased the total protein content in the cell cultures (Figure 6, Caco-2: TPC).

To sum up, the results obtained for all employed assays show that silver complexes and silver salts evoked concentration-dependent decreases in cell viability. MTZ in the used concentration range did not affect cell viability (Figure 6). The analysis of IC_50_ values revealed that [Ag(MTZ)_2_]_2_SO_4_ is more cytotoxic toward cancer cells HepG2 and Caco-2 than Ag_2_SO_4_. Interestingly, both [Ag(MTZ)_2_]_2_SO_4_ and Ag_2_SO_4_ presented similar levels of cytotoxicity against normal mice fibroblasts (Balb/c 3T3). Reversely for cancer cells, silver(I) nitrate caused cell death more efficiently than its complex. The cytotoxicity of AgNO_3_ and [Ag(MTZ)_2_NO_3_] were comparable for non-cancer cells. Our study confirms that Ag(I) complexes show selective cytotoxicity toward various types of cells, and this is dependent on the type of ligand linked to the silver(I) ions. This dependency is probably connected with the stability of the complexes and the hydrophilicity–lipophilicity of the complexes formed by the type of the ligand. Metronidazole is a well-known antibiotic drug used to treat certain parasitic and bacterial infections. Metronidazole suffers a ferredoxin-mediated reduction, forming reactive oxygen species capable of damaging DNA. This results in the genotoxic activity of the drug, which is not always associated with a decrease in cell viability. On the other hand, combining the drug with silver ions increases the cytotoxicity of the complexes compared to metronidazole. The mechanism of cytotoxic action of the complexes depends on many factors. This was shown in this study with the substituent of the silver ion and with the type of cells.

### 3.5. Antimicrobial Activity

The results of the antibacterial and antifungal activities are presented in Table 4. Metronidazole and silver sulfadiazine were used as the standard drugs. The tested complexes revealed antimicrobial activity comparable to Ag sulfadiazine. Notably, the complexes showed increased bioactivity against aerobic and facultative anaerobic, Gram-positive and Gram-negative bacteria when compared to metronidazole. Here, [Ag(MTZ)_2_NO_3_] and [Ag(MTZ)_2_]_2_ SO_4_·5H_2_O showed, similar biological activity against the majority of tested bacterial strains with a minimal inhibition concentration (MIC) in the range of 0.24–250 mg/L. The MIC values for reference bacteria indicated very strong (MIC < 10 mg/L) or strong (MIC 10–25 mg/L) anti-staphylococcal activity (*S. aureus*, *S. epidermidis*), anti-micrococcal activity (*M. luteus*) and activity against sporogenic bacilli (*B. subtilis*) with the tested compounds [29]. They had also very strong bioactivity against all tested strict anaerobic Gram-positive (*A. israelii, C. perfringens*) and Gram-negative (*B. fragilis*, *P. intermedia*, *F. nucleatum*, *V. parvula*) bacteria. Strong bioactivity was observed for Gram-negative aerobic and facultatively anaerobic bacteria for both compounds. Good bioactivity (MIC 26–125 mg/L) of the tested complexes was found against enterococci, sporogenic bacilli *B. cereus* and strict anaerobic *Propionibacterium acnes*. The least sensitive strain was *Campylobacter jejunii*, with an MIC of 126–500 mg/L for [Ag(MTZ)_2_]_2_SO_4_·5H_2_O, showing moderate antibacterial activity. The low values of the MBC/MIC ratio (1–4) for the complexes suggested their bactericidal power against all tested references strains. UV exposition has no impact on the antibacterial activity of tested Ag complexes.

### 3.6. Quantitative and Qualitative Determination of Antibiofilm Activity

In the quantitative CV method, biofilm eradication at 50% was observed for both tested silver complexes with metronidazole at a concentration of 125 mg/L against *S. aureus* biofilm. *E. coli* biofilm appeared to be less sensitive, and 1000 mg/L only eradicated biofilm on the level proximate to 50% (Figure 7). In the CLSM method, the tested compounds showed the ability to inhibit biofilm formation only at higher tested concentrations (equal or above 125 mg/L). Moreover, better effectiveness was observed for Gram-positive bacteria (*S. aureus*) compared to Gram-negative ones (*E. coli*), which is consistent with the results obtained with quantitative crystal violet assay (Figure 8). Both tested complexes ([Ag(MTZ)_2_]_2_SO_4_·5H_2_O and [Ag(MTZ)_2_NO_3_]) at a concentration of 125 mg/L only slightly inhibited the growth of *S. aureus* biofilm (Figure 8a). The biofilm structure and density were comparable to the control; however, noticeably more dead cells were observed, resulting in a more yellowish color of the biofilm. Both compounds applied at a concentration of 500 mg/L significantly reduced biofilm density. Importantly, the obtained effect was similar to the effect provided by 500 µg/mL of the reference drug, Ag sulfadiazine. Nevertheless, unlike the tested complexes, Ag sulfadiazine at a concentration of 125 mg/L had the ability to meaningfully inhibit *S. aureus* biofilm growth. 

Interestingly, control biofilm formed by *E. coli* had an extremely dense exopolymeric matrix that hindered the penetration of the DNA probes, making it impossible to stain the biofilm structure. Thus, a Nomarski contrast was additionally applied to visualize the control *E. coli* biofilm (Figure 8b). Importantly, the higher the concentration of tested compounds applied, the better the observed biofilm staining was, indicating that complexes had the ability to disturb the biofilm structure, enabling DNA probe penetration and fluorescent staining. Low concentrations of tested compounds (<125 mg/L) were ineffective or only slightly disturbed the biofilm structure. Higher concentrations (125 mg/L and 500 mg/L) of the complexes had better effectiveness than Ag sulfadiazine, since biofilm contained more dead cells and exhibited a more yellowish color. This observation confirmed the results obtained with the crystal violet test that showed higher antibiofilm activity of novel compounds compared to the reference drug.

## 4. Discussion

Scientists are still looking for new chemical compounds with desired biological properties by modifying the structure of molecules known for a long time or creating new adducts from substances with well-documented activity. Many of the previous studies have shown the chemical and biological properties of the coordination compounds of imidazole derivatives and transition metals, including silver [30]. The majority of studies involve the chemical and biological properties of [Ag(MTZ)_2_NO_3_]. However, [Ag(MTZ)_2_]_2_SO_4_ has not been explored due to the fact that Ag_2_SO_4_ is poorly soluble in water and other solvents. Therefore, we decided to synthetize and determine the chemical properties of the novel complex, including its solubility. We found that [Ag(MTZ)_2_]_2_SO_4_ is more soluble in water in comparison to Ag_2_SO_4_; thus, in the next step of our study, we evaluated its microbiological activity and cytotoxicity. For these reasons, the aim of our study was to compare the chemical, cytotoxic and microbiological properties of metronidazole complexes with silver(I) nitrate and silver(I) sulfate to metronidazole and pure silver(I) salts.

The important feature of potential biologically active compounds is their stability. Our study revealed that the studied complexes [Ag(MTZ)_2_NO_3_] and [Ag(MTZ)_2_]_2_SO_4_ exhibited similar photostability both in the daylight and UV-A, with similar solubility in water. Our recent studies utilizing various environmental conditions (temperature, pH, the presence of an oxidizing agent) confirmed that complexes showed higher stability than pure salts [31]. Moreover, our one-step process of [Ag(MTZ)_2_]_2_SO_4_ synthesis, developed for the first time, is effective, easy and delivers a high-quality product. 

These results allowed us to perform the evaluation of biological properties, namely, cytotoxicity toward human cancer cell lines and normal mice fibroblasts, as well as microbiological activity.

The fundamental stage of the preclinical studies of substances that could become drugs in the future is the cytotoxicity tests [32,33,34]. Cytotoxicity studies provide basic information about mechanism of action of the tested compound as a potential drug [20,23,35,36]. For the preliminary evaluation of anti-cancer potential of the tested complexes, we employed mice normal cells (Balb/c 3T3) and human cancer cell lines (HepG2, Caco-2). The HepG2 cell line is widely used as a parameter to predict toxicity [37]. In our study, we also introduce the Caco-2 cell line, the cells of which divide spontaneously in culture to form monolayers of mature intestinal enterocytes [38]. We have used them as an intestinal barrier model in in vitro toxicology studies. The obtained results clearly indicate that the combination of metronidazole with Ag_2_SO_4_ exerts higher cytotoxicity toward cancer cells than the combination of metronidazole with AgNO_3_. Reversely, [Ag(MTZ)_2_NO_3_] induced normal cell death more effectively than [Ag(MTZ)_2_]_2_SO_4_. Comparing the sensitivity of normal cells to cancer cells for the studied complexes, we found that normal cells were more sensitive to [Ag(MTZ)_2_NO_3_], whereas cancer cells were more sensitive to [Ag(MTZ)_2_]_2_SO_4_. These observations may suggest that the differences in the cytotoxic potential of studied complexes can be affected by different counter-ions. However, we reported a higher cytotoxicity of AgNO_3_ than Ag_2_SO_4_ in the case of cancer cells, and this was comparable in the case of fibroblasts. To the best of our knowledge, there is no other study comparing the cytotoxicity of silver (I) complexes containing MTZ as a ligand and different counter-ions. We found only one study performed by Stryjska et al., who compared the cytotoxic potential in HepG2 cells and fibroblasts of silver(I) complexes containing miconazole (MCZ) as a ligand and AgNO_3_ or AgClO_4_. The results of MTT, NRU, LDH and TPC assays displayed that these two complexes did not differ significantly. The observation suggests that various counter-ions did not differentiate cytotoxicity. In turn, the efficiency in inducing HepG2 cell and fibroblast death was various for complexes with the same silver(I) salt (AgNO_3_), but different ligands. In addition, [Ag(MCZ)_2_NO_3_] showed a better efficiency of elimination with HepG2 cancer cells and lower toxicity to Balb/c 3T3 fibroblasts for [Ag(MCZ)_2_NO_3_] in relation to the silver(I) complex of metronidazole [Ag(MTZ)_2_NO_3_]. Moreover, the complex of MCZ with AgNO_3_ and MTZ with AgNO_3_ were more toxic to HepG2 cancer cells in comparison to the free ligands [7]. Our previous study demonstrated that HepG2 cells exhibited similar sensitivity to [Ag(4-OHMePy)_2_]NO_3_ and [Ag(MTZ)_2_NO_3_], whereas mice fibroblasts generally were more sensitive to [Ag(MTZ)_2_NO_3_]. Interestingly, in human pancreatic cancer cells, PANC-1 and 1.2B4, we found that [Ag(MTZ)_2_NO_3_] and [Ag(4-OHMePy)_2_]NO_3_ had similar cytotoxic potential. On the other hand, these results may suggest that the incorporation of the same silver(I) salt to different ligands did not differentiate the cytotoxic activity [4]. Azoles are very interesting compounds due to their in vitro cytotoxic properties [32,39,40,41]. Their properties can be regulated by combining them with various ligands [4,10,34,42].

The anticancer properties of Ag(I) ions are still being extensively studied in multiple types of human cancer cells, such as MCF-7 breast cancer, lung cancer cells, H1299, hepatocarcinoma cells HepG2 and pancreatic cancer cell PANC-1. The majority of data gathered for Ag(I) ions are derived from studies on silver(I) nitrate and silver nanoparticles. It was demonstrated that the anticancer properties of Ag(I) ions involved several mechanisms, including (1) the generation of reactive oxygen species (ROS) that destroy the mitochondrial respiratory chain [43,44], (2) causing DNA damage [3,45], (3) the induction of cell apoptosis, necrosis and autophagy [46,47,48,49], (4) the inhibition of cell migration and invasion [47] and (5) the induction of lactate dehydrogenase leakage (LDH) [50]. So far, only limited biochemical research in antibacterial studies of silver(I) complexes was conducted to tackle the fundamental issue of how silver(I) ions, as well as nitrate or sulfate anions, enter cells. Recent studies have revealed that another silver(I) salt, namely sodium silver(I) thiosulfate complex (STS), also exerted anticancer activity involving dose-dependent cytotoxicity toward MCF-7 breast cancer cells and myeloid leukemia cells K562 [51]. STS induced cancer cell death via ROS generation, cell cycle arrest at the G1 phase, a decrease in the GSH levels and morphological changes, such as the expansion of cell shapes, cell flattening and the loosening of cell–cell contacts [51]. 

Infections caused by anaerobic bacteria can be serious and life-threatening and occur frequently. Anaerobes colonize the skin, and are the main component of mucous membranes microbiota, so they are endogenous in nature [52]. The treatment of anaerobic infection is complicated due to their polymicrobial nature. Anaerobes generally are isolated and mixed with aerobes. Then, the chosen antimicrobial should cover a spectrum of both groups of pathogens. The most effective antimicrobials against anaerobes are metronidazole, the carbapenems, chloramphenicol, the combinations of penicillin and beta-lactamase inhibitors, tigecycline, cefoxitin and clindamycin [52]. However, the widespread use, overuse and misuse of antimicrobials during the last 80 years have been associated with the explosion of antimicrobial resistance [53]. Antimicrobial strategies able to overcome the development of resistance and innovative perspectives on the new approach are immediately needed. We showed a broad range of metronidazole complex activity by combining it with silver salts. The complexes tested in our study potentially offer additional options for therapy in infections caused by mixed pathogens that are Gram-positive, Gram-negative, aerobic and anaerobic, or resistant to conventional antibiotic therapy. 

The higher cytotoxic and antimicrobial activities of the studied silver(I) complexes, in comparison to pure silver(I) salt or free ligands, may be connected with their chemical properties. Numerous studies demonstrated that silver(I) complexes present higher bioavailability and the prolonged release of silver or gold ions [54,55,56,57,58,59,60]. The formation of new chemical bonds between the Ag–N-heterocycle slow metal release, depending on the degradation and/or redox processes of the complex. As a result, the stability and half-life of silver(I) complexes with N-heterocycle ligands significantly increases [61] Consequently, the release of active Ag(I) ions from the complexes follows gradually and longer, enhancing the interaction with cellular (eukaryotic, bacterial and mitochondrial) membranes, enzymes and DNA/RNA [30]. Dibrov et al. [62] found out that the addition of low micromolar concentrations of Ag(I) ions to inside-out membrane vesicles of *V. cholerae* induced a total collapse of both respiration-generated transmembrane pH gradient (ΔpH) and the respiration-generated membrane electric potential Δψ, irrespective of the presence of Na^+^ ions. The interaction between silver and the constituents of the bacterial membrane causes structural changes and damage to the membranes, and intracellular metabolic activity is one of the most important mechanisms of Ag(I) ion toxicity, which might be the cause or consequence of cell death [63]. The antibacterial mode of action of silver ions was nicely reviewed by Kędziora et al. [64], and are connected with interactions with the bacterial cell envelope, interactions with molecules inside the cell (e.g., nucleic acids and enzymes) and the production of reactive oxygen species (ROS). Differences between the mode of Ag(I) ion action against Gram-positive and Gram-negative bacteria may rely on the means of silver uptake into the cell. Silver ions enter Gram-negative cells via the major outer membrane proteins (OMPs); however, it has been widely discussed, and the results of the conducted experiments were quite different [64]. Such actions display the improvement of the efficacy of Ag(I) complexes in increasing cytotoxicity toward cancer and normal cells and antibacterial activity. In addition, it was found that Ag(I) complexes gain new biological properties, namely antimycotic, since neither silver(I) salts nor some ligands have this property, or they have it, but it is worse [9,65,66,67]. 

In our previous study we investigated the antibacterial and antimycotic properties of metronidazole complexes with silver(I) salts bearing various counter-ions. We found that silver(I) complexes with metronidazole exhibited not just antibacterial, but also antimycotic activity. What is more, the type of counter-ion exerted a significant impact on the efficacy of the antimicrobial activity [12].

## 5. Conclusions

The newly synthetized [Ag(MTZ)_2_]_2_SO_4_·5H_2_O complex exhibits a polymeric structure, in contrast to [Ag(MTZ)_2_NO_3_]. In the ^13^C NMR spectra of the novel MTZ complex, we found that the greatest shift, of about 1.34 ppm, was observed for the methyl group. Both complexes of metronidazole with silver(I) salts show greater stability in daylight and after exposure to UV-A radiation, as well as solubility in water, in comparison with the solubility of Ag_2_SO_4_ in water or MTZ in water. This is due to the greater stability of the complex cation, in which silver(I) is bound to the ligand, and not in the free form. Generally, the obtained results showed that the complexes exhibit better biological properties than pure silver(I) salts or MTZ. In the case of *E. coli*, the biofilm inhibition effect by the silver coordination compounds is more pronounced in relation to the reference drug. Of note, we used, for the first time, Ag_2_SO_4_, which because of its undesired chemical properties had not been utilized so far, to perform the synthesis of a novel complex, and we thereby obtained a compound with the desired chemical and biological properties. The probable synergic mechanism of action of both complex components—DNA damage by metronidazole and the binding to thiol groups (-SH) of proteins from silver compounds—is responsible for the antitumor and bactericidal activities of the tested complexes. Thus, [Ag(MTZ)_2_]_2_SO_4_ and [Ag(MTZ)_2_NO_3_] can be regarded as a compounds that exhibit a unique “risk–benefit” relationship, based on the low cytotoxicity properties to fibroblasts (Balb/c 3T3) and their antimicrobial activity.

Taken together, our in vitro preliminary studies provided the background for further research on the potential use of the studied complexes as pharmaceutical agents, which we intend to continue. After all, the metronidazole silver(I) complex of the formula [Ag(MTZ)_2_NO_3_] has already been tested in clinics as part of a medical experiment [68], and is now used as a prescription drug prepared in a pharmacy. The formulations of drug forms, i.e., eye ointment and drops and ointment for external use, have been developed for the treatment of patients.

## Data Availability

The data presented in this study are available in this article and the Appendix A.

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
