# Peer review of "Multifunctional Silver(I) Complexes with Metronidazole Drug Reveal Antimicrobial Properties and Antitumor Activity against Human Hepatoma and Colorectal Adenocarcinoma Cells"

_cancers, 2022, doi:10.3390/cancers14040900_

Round 1

Reviewer 1 Report

The authors Żyro et al. revised their manuscript ‘Multifunctional Silver(I) Complexes With Metronidazole Drug Reveal Antimicrobial Properties and Antitumor Activity Against Human Hepatoma and Colorectal Adenocarcinoma Cells’ to comply with my comments and suggestions. Therefore, I recommend publication of the manuscript.

Author Response

Dear Rewiever,

Sincerely Yours,

Prof. dr. Justyn Ochocki

Reviewer 2 Report

The last modifications made by the authors provided a better clarification of some aspects in the antiproliferative activity of Ag(I) complexes. However, it still remains the doubt about why or how counter-ions, as sulfate, nitrate, thiosulfate can act possibly as adjuvants in this process.

I suggest:

 in page 20,  line 624 - to change "formation of DNA damage' to "causing DNA damage"

in page 20, line 626 - to include a phrase: "So far, only limited biochemical research in antibacterial studies of silver(I) complexes was done to tackle the fundamental issue of how silver(I) ions as well as nitrate or sulfate anions enter cells."

I also strongly recommend that the insertion of the different silver compounds in the studied tumor cells be determined in future more significant studies.

Therefore, in the last revised version, the manuscript can be accepted.

Author Response

Dear Rewiever,

Sincerely Yours,

Prof. dr. Justyn Ochocki

This manuscript is a resubmission of an earlier submission. The following is a list of the peer review reports and author responses from that submission.

Round 1

Reviewer 1 Report

This is a revisited study on silver(I) metronidazole complexes by the authors. A previous study describes the antimicrobial activity of [Ag(MTZ)2X] complexes, [MTZ = 1-(2-hydroxyethyl)-2-methyl-5-nitro-1H-imidazole (metronidazole drug), with different counter-ions, [X = NO3−, ClO4−, CF3COO−, BF4− and CH3SO3−]. In this new manuscript the antitumor activity of two compounds, [Ag(MTZ)2NO3] and [[Ag(MTZ)2]2SO4  is also investigated.

Some points deserve more discussion.

Although in the introduction the authors describe that the antimicrobial activity of silver compounds is based on the binding to thiol groups (‐SH) of proteins, no mention was made about their possible mechanisms of action as cytotoxic agents toward tumor cells. What is the rule of ligands in their reactivity, both antimicrobial and antitumor? Particularly, what is the rule of metronidazole?

It is difficult to understand why “In human cancer cells [Ag(MTZ)2]2SO4 was more cytotoxic than Ag2SO4, in turn AgNO3 was more cytotoxic than [Ag(MTZ)2NO3]. For Balb/c 3T3 cells Ag2SO4 was more cytotoxic than [Ag(MTZ)2]2SO4, while the cytotoxicity of AgNO3 and [Ag(MTZ)2NO3] was similar.” In solution, the counter-ions usually dissociate, and possibly differences could be explained by solubility. How these complexes enter the cells? As cations?

I suggest monitoring the insertion of silver in the cells (metal concentration determined by ICP-MS, for instance), and to correlate these concentrations to the corresponding cytotoxicity.

Metronidazole, on the other hand, suffers a ferrodoxin-mediated reduction (nitro group reduced to amine group), possibly forming reactive species (radicals?) capable of damaging DNA. However, in the data described it has no effect in the viability of the cells. When coordinated to silver ions, this reaction is also observed?

 Another point is the stability of the complexes under light. Apparently the ligand metronidazole increases the photo-stability of silver, retarding the liberation of Ag+ ions, and  its reduction to silver(0).          ...” the studied complexes [Ag(MTZ)2NO3] and [Ag(MTZ)2]2SO4    exhibited similar photostability both in the day‐light and UV‐A and solubility in water”.  This could explain a lower reactivity of the complexes regarding the silver salts.

Different targets were monitored in the described studies (mitochondria, lysosome, cellular membrane). A better discussion/interpretation of these results would be interesting, comparing them.

Biofilm inhibition effect (%) of the complexes are also not remarkably different, considering experimental errors (Fig. 7).

Considering the large amount of data, conclusions are very poor. Stronger arguments should be used to differentiate the complexes and corresponding silver salts, correlating all the results by different techniques. Additionally, some considerations about the mode of action of these studied complexes as potential antitumor or microbicidal agents should be added.

Author Response

Dear Rewiever #1,

Reviewer 2 Report

The manuscript by Żyro et al. entitled “Multifunctional Silver(I) Complexes With Metronidazole Drug Reveal Antimicrobial Properties and Antitumor Activity Against Human Hepatoma and Colorectal Adenocarcinoma Cells” assesses the effect of the sulfate counter-ion in a metronidazole-silver (I) complex, [Ag(MTZ)2]2SO4, as opposed to the previously reported [Ag(MTZ)2]NO3, on the structure, photostability, antimicrobial and anticancer activities.

Generally, the manuscript is well written, the results are well presented and discussed, and the paper lies on current topics, namely anticancer and antimicrobial research. There are, however, some issues that need to be addressed:

  • While I agree that [Ag(MTZ)2]2SO4 has some different structural and biological properties in comparison to its counterpart, [Ag(MTZ)2]NO3, it cannot be considered a ‘new’ compound/complex. Therefore, I strongly believe that the authors should avoid using the term ‘new’.
  • Line 257: ‘statistically significant at p ≤ 0.05.5’ – please correct this error.
  • Minor English mistakes are found throughout the entire manuscript. For instance, the authors repeatedly use ‘the study compounds’, which should be replaced with ‘the studied compounds’.
  • Section 3.4. Cytotoxicity of the Compounds should be rewritten; it should be made clearer and easier to read.
  • In the conclusions section, the authors should explain in more detail why ‘the complexes exhibit better biological properties than pure silver(I) salts or MTZ’. Also, lines 633-635: ‘13C NMR spectra of novel MTZ complex were found to be nearly identical like for free ligand. Upon metal coordination, the signals of carbon atoms are barely shifted. The greatest shift, of about 1.34 ppm, was observed for the methyl group’ – I believe this information can be deleted or shortened.

In conclusion, I recommend minor revisions prior to the publication of this manuscript in Cancers.

Author Response

Dear Rewiever #2,

Round 2

Reviewer 1 Report

The main point is the insertion of the silver compounds in the cells or microbes, and this was not clarified. Also, a  good explanation about differences between the nitrate and sulfate counter-ions was not provided. What are their rules in the anticancer or antimicrobial activities? So, I think the interest by the readers and overall merit of the work is limited.

Author Response

Dear Reviewer,

thank you very much for your valuable comments.

Sincerely yours,

Prof. dr. Justyn Ochocki
